# PREFERENCE-AWARE MIXTURE-OF-EXPERTS FOR MULTI-OBJECTIVE COMBINATORIAL OPTIMIZATION

## ABSTRACT

Recent neural methods for multi-objective combinatorial optimization involve solving preference-specific subproblems with a single model and have achieved competitive performance. However, they still suffer from limited learning efficiency and insufficient exploration of the solution space. This paper conducts a theoretical analysis that reveals the equivalence between this single-model paradigm and an implicit Mixture-of-Experts architecture. Furthermore, we propose a Preference-Aware mixture-of-experts (PA-MoE) framework that learns preference-specific representations while explicitly modeling *preference-instance* interactions. By integrating a sparsely activated expert module with an innovative preference-aware gating mechanism, PA-MoE enhances preference-conditioned representation learning while preserving parameter efficiency. Moreover, PA-MoE is generic and can be applied to three different neural MOCO solvers. Experimental results on the multi-objective traveling salesman problem (MOTSP), multi-objective capacitated vehicle routing problem (MOCVRP), and multi-objective knapsack problem (MOKP) show that PA-MoE is able to generate a Pareto front with higher diversity, achieving superior overall performance.

## 1 INTRODUCTION

Multi-objective combinatorial optimization (MOCO) involves solving problems with multiple, often conflicting objectives, where the goal is to approximate a diverse Pareto front that reflects trade-offs between objectives Lust & Teghem (2010); Liu et al. (2020). MOCO inherits NP-hardness from its underlying combinatorial structure, and the multi-objective aspect further compounds its intractability. Although heuristic MOCO solvers have long provided competitive performance within reasonable runtimes Jozefowiez et al. (2008); Florios & Mavrotas (2014); Ehrgott et al. (2016), recent advances in deep reinforcement learning (DRL) provide a neural, data-driven alternative, enabling scalable policies and reducing dependence on handcrafted design choices Zhang et al. (2021); Shao et al. (2021); Zhang et al. (2023).

A *decomposition* scheme Navon et al. (2021); Lin et al. (2024) is widely adopted in DRL-based MOCO solvers, where the MOCO problem is scalarized into preference-weighted subproblems, each tied to a specific preference, and the resulting subproblems are solved end-to-end with deep models. Within this paradigm, the *single-model* method trains a single preference-conditioned neural network (usually using a Transformer Vaswani et al. (2017) backbone) to solve all subproblems. Its effectiveness has been demonstrated by several recent works that achieve state-of-the-art (SOTA) performance across multiple MOCO benchmarks Lin et al. (2022); Chen et al. (2023); Fan et al. (2024); Chen et al. (2025).

Although empirical results are promising, current works mainly focus on *single-model* architectural refinements (e.g. alternative *preference-instance* interactions and their placement within the model), and the theoretical foundations of their effectiveness remain underexplored. In response, we first reinterpret preference-conditioned Transformers as *implicit* Mixture-of-Experts (MoE) models, where attention layers act as dense routers over token-indexed affine experts. However, in this view, we observe that current *single-model* approaches lack controllable sparsity and explicit specialization, which inevitably results in insufficient representation learning and reduced sample efficiency under scalarized training paradigm in MOCO.

Subsequently, we introduce an explicit Preference-Aware MoE (PA-MoE) framework. PA-MoE replaces feed-forward layers with sparsely activated expert modules and employs a lightweight gating mechanism conditioned on both instance structure and preference. This design delivers significant performance improvements for neural MOCO solvers, without compromising parameter efficiency. Our contributions can be summarized as follows:

- We offer a novel insight by interpreting unified preference-conditioned Transformers for MOCO as *implicit* Mixture-of-Experts models, providing a fresh view on the connection between existing MOCO methods and modular expert architectures.
- Driven by this perspective, we propose a PA-MoE framework that explicitly employs sparsely activated expert modules and an innovative lightweight gating mechanism. PA-MoE enhances model capacity and enables deeper exploration of the solution space, while avoiding inefficient representation learning that often leads to suboptimal performance.
- The proposed PA-MoE framework is modular and versatile, enabling architecture-agnostic integration into three representative neural MOCO solvers. Extensive experiments demonstrate the significant superiority of our approach over SOTA neural methods.

## 2 RELATED WORK

**Traditional Methods for MOCO.** Exact algorithms for MOCO are capable of identifying all Pareto-optimal solutions through exhaustive enumeration Ehrgott et al. (2016); Bergman et al. (2022) . In practice, multi-objective evolutionary algorithms (MOEAs) are widely used to approximate the Pareto front under limited computational resources Tian et al. (2021); Xie et al. (2022); Lin et al. (2021); García-Martínez et al. (2007). Algorithms such as dominance-based MOEAs Deb et al. (2002) and decomposition-based MOEAs Zhang & Li (2007); Ke et al. (2014) often incorporate local search or domain-specific heuristics to improve solution quality. However, these methods typically require extensive manual design and fine-tuning of evolutionary operators, such as crossover and mutation, which limits their flexibility and scalability Fang et al. (2020).

**DRL-based methods for MOCO.** DRL-based methods have recently gained attention due to their fast inference and easy deployment across various MOCO problems. Early efforts focused on training separate policy networks for each scalarized subproblems, termed the *multi-model paradigm* Li et al. (2021); Wu et al. (2020); Shao et al. (2021); Zhang et al. (2021). For example, Meta-DRL (MDRL) Zhang et al. (2023) adapts a pre-trained meta-model through fine-tuning to accommodate varying preferences. However, training dedicated models for each subproblem introduces substantial training overhead and the burden of maintaining multiple networks. Alternatively, the *single-model paradigm* has been introduced, wherein a unified model is trained to solve subproblems for arbitrary preference vectors. For example, given a preference as input, PMOCO Lin et al. (2022) employs a hypernetwork to generate the decoder parameters for each subproblem. Beyond decoder-level adaptation, some methods (e.g., CNH Fan et al. (2024)) enhance *preference-instance* interaction using dual-attention in the encoder, while others (e.g., WE-CA Chen et al. (2025)) directly modulate node embeddings via feature-wise affine transformations for better alignment with trade-off preferences.

## 3 PRELIMINARIES

### 3.1 MULTI-OBJECTIVE COMBINATORIAL OPTIMIZATION (MOCO)

Given $M$ objectives, the MOCO problem can be defined as:

$$\min_{x \in X} f(x) = \big(f_1(x), f_2(x), \ldots, f_M(x)\big), \tag{1}$$

where $X$ is a discrete decision space and $f : X \to \mathbb{R}^M$ maps each solution $x \in X$ to an $M$-dimensional objective vector. A solution $x_1 \in X$ is said to *Pareto-dominate* another solution $x_2 \in X$ (denoted $x_1 \prec x_2$) if $f_i(x_1) \leq f_i(x_2)$ for all $i \in \{1, \ldots, M\}$, and there exists at least one objective $j \in \{1, \ldots, M\}$ such that $f_j(x_1) < f_j(x_2)$. A solution $x^* \in X$ is *Pareto optimal* if there is no other solution $x' \in X$ such that $x' \prec x^*$. The set of all Pareto optimal solutions, known as *Pareto set*, is defined by $P = \{ x^* \in X \mid \nexists x' \in X : x' \prec x^* \}$. The set of their corresponding objective vectors constitutes the *Pareto front*, formally given by $F = \{ f(x) \in \mathbb{R}^M \mid x \in P \}$.

### 3.2 Subproblem Formulation

In this paper, we focus on the decomposition-based paradigm of MOEA/D, which partitions a MOCO problem into multiple subproblems, each modeled as a Markov Decision Process (MDP). Given the MDP formulation, the policy network is trained using the REINFORCE algorithm Williams (1992), which maximizes the expected reward over the sampled trajectories. The objective in DRL can be optimized with the corresponding policy gradient:

$$\nabla_\theta \mathcal{L}(\theta \mid \mathcal{I}) = \mathbb{E}_{\pi \sim P_\theta(\cdot \mid \mathcal{I})} \left[ (R(\pi) - b(\mathcal{I})) \nabla_\theta \log P_\theta(\pi \mid \mathcal{I}) \right], \tag{2}$$

where $b(\mathcal{I})$ denotes a baseline function used to reduce gradient variance and stabilise learning across varying instances $\mathcal{I} \in \tilde{\mathcal{I}}$. The trained autoregressive neural policy can be used to incrementally construct solutions for subproblems.

### 3.3 Single-Model Paradigm for MOCO Subproblems

The *single-model* paradigm trains a single parameterized solver $\pi_\theta$ that maps an instance $s$ and a preference vector $\lambda$ directly to a solution of the corresponding scalarized subproblem. Under the above DRL formulation, the current paradigm *single-model* employs preference–instance interaction to fuse preference and instance information, implemented in the encoder or the decoder of a shared Transformer Vaswani et al. (2017).

As shown in Figure 1 (a), the early *preference-instance* interaction strategy operates in the decoder layers. The preference $\lambda$ is input to a hypernetwork that dynamically generates the attention parameters of the decoder. These preference-specific parameters are subsequently used to compute attention over the node embeddings produced by the encoder, enabling the decoder to output a distribution over the remaining feasible nodes at each decoding step. In this way, it ensures that solution construction remains conditioned on the given preference throughout the generation process.

Afterward, some works introduce the *preference-instance* interaction at the encoder layers. As illustrated in Figure 1 (b), the preference $\lambda$ is used to modulate the node representations within the encoder, enabling the model to learn preference-aware embeddings prior to the decoding phase. This modulation is typically performed before the skip connection and instance normalization He et al. (2016); Ulyanov et al. (2016), and can be performed through preference-aware attention mechanisms. Common designs include FiLM-style conditioning Perez et al. (2018), dual attention-based *preference-instance* conditioning Fan et al. (2024) that generate modulation parameters from preferences.

## 4 Methodology

### 4.1 Theory Basis: From Implicit MoE to Explicit MoE

Self-attention is the core component of Transformer architectures Vaswani et al. (2017). Therefore, the theoretical analysis in this section begins from the perspective of the self-attention mechanism.

**Preference-conditioned Attention Meet Implicit MoE.** A unified preference-conditioned attention can be viewed as an *implicit* MoE Shazeer et al. (2017). By conditioning $Q/K/V$ on the preference $\lambda$, attention effectively behaves as *soft routing* over shared capacity, enabling conditional specialization across subproblems without parameter duplication.

Let $X = [x_1^\top, \ldots, x_N^\top]^\top \in \mathbb{R}^{N \times d}$ be node embeddings of instance $s$. For the $l-th$ head in the attention layer, the projections with preference conditioning can be defined as $Q^{(l)}(\lambda) = X W_Q^{(l)}(\lambda) \in \mathbb{R}^{N \times d_k}$, $K^{(l)}(\lambda) = X W_K^{(l)}(\lambda) \in \mathbb{R}^{N \times d_k}$, $V^{(l)}(\lambda) = X W_V^{(l)}(\lambda) \in \mathbb{R}^{N \times d_v}$, respectively.

Then, given $i$ and $\forall j$, let $q_i^{(l)}(\lambda)$ and $k_j^{(l)}(\lambda)$ be the $i$-th and $j$-th row of $Q^{(l)}(\lambda)$ and $K^{(l)}(\lambda)$, respectively, and $v_j^{(l)}(\lambda)$ be the $j$-th row of $V^{(l)}(\lambda)$. The head output at position $i$ is:

$$h_i^{(l)}(X, \lambda) = \sum_{j=1}^N \alpha_{ij}^{(l)}(X, \lambda)\, v_j^{(l)}(\lambda), \quad \alpha_{ij}^{(l)}(X, \lambda) = \frac{\exp\left(\frac{q_i^{(l)}(\lambda)\, k_j^{(l)}(\lambda)^\top}{\sqrt{d_k}}\right)}{\sum_{k=1}^N \exp\left(\frac{q_i^{(l)}(\lambda)\, k_k^{(l)}(\lambda)^\top}{\sqrt{d_k}}\right)}. \tag{3}$$

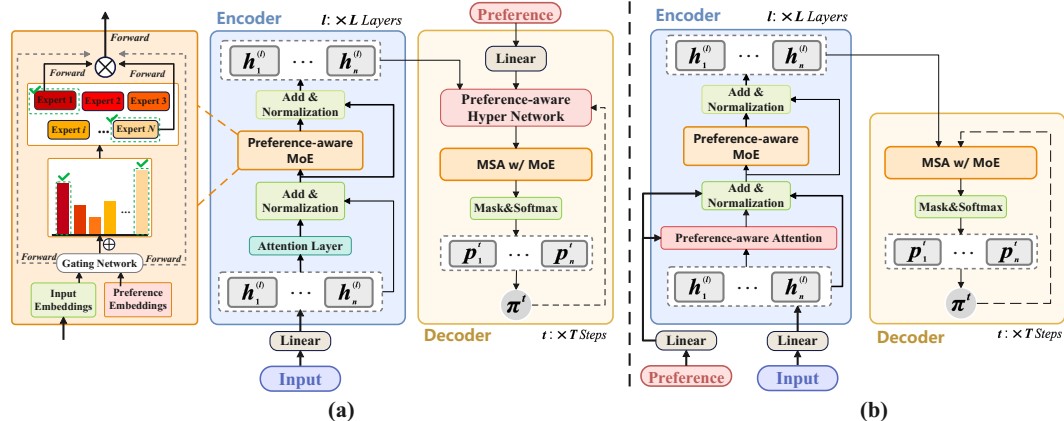

Figure 1: Neural architectures of PA-MoE under two *preference-instance* interaction strategies. Left: Decoder-side interaction with a preference-aware hypernetwork generating parameters for attention layers Lin et al. (2022). Right: Encoder-side interaction with the preference-aware attention mechanism Fan et al. (2024). In both, the proposed MoE adapts the policy learning dynamically based on instance and preference.

Next, the output of the $l$-th head is:

$$h_i^{(l)}(X, \lambda) = \sum_{j=1}^{N} \underbrace{\text{softmax}_j\big(s_{ij}^{(l)}(X, \lambda)\big)}_{\text{preference-aware routing}} \quad \underbrace{\mathcal{E}_j^{(l)}(X, \lambda)}_{\text{shared (affine) experts}}, \quad (4)$$

where the *experts* are defined as $\mathcal{E}_j^{(l)}(X, \lambda) := v_j^{(l)}(\lambda)$ and the *gating scores* as $s_{ij}^{(l)}(X, \lambda) :=$ $\frac{q_i^{(l)}(\lambda) \, k_j^{(l)}(\lambda)^\top}{\sqrt{d_k}}$. Thus, each attention head is an *implicit* MoE that mixes $N$ experts with a gating distribution that depends on the preference $\lambda$ via $W_Q^{(l)}(\lambda), W_K^{(l)}(\lambda)$ (and optionally $W_V^{(l)}(\lambda)$).

Inspired by the above theoretical analysis, we discuss the rationale and novelty of our proposed preference-aware MoE architecture subsequently.

**Preference-Aware MoE.** While preference-conditioned attention realizes an *implicit* MoE, the "experts" are token-indexed linear maps tied to the value projection, and the softmax router is dense. As a result, the sparsity of MoE and the utilization of experts cannot be explicitly controlled. In other words, the computation cannot be selectively focused on structures that most decrease the scalarized objective $g(f(\pi \mid I), \lambda)$, especially under wide preference spans or strongly conflicting objectives.

Based on the above analysis, we introduce an *explicit* Preference-Aware MoE (PA-MoE) with non-linear expert subnetworks and a router $w_k(h, \lambda)$ that depends jointly on the hidden state and preference. Replacing them with *explicit* non-linear experts $\{E_k\}_{k=1}^m$ and a router $w_k(h, \lambda)$ yields the hypothesis class as:

$$\mathcal{H}_{\text{PA-MoE}} = \left\{ h \mapsto \sum_{k=1}^{m} w_k(h, \lambda) \, E_k(h) \,\Big|\, h \in \mathbb{R}^{d_h}, \, w_k(h, \lambda) \geq 0, \, \sum_k w_k(h, \lambda) = 1 \right\}, \quad (5)$$

which (under mild conditions) strictly enlarges the implicit attention class and supports controllable sparsity (using Top-$K$ activation) and load balancing via an auxiliary loss.

By default, the above theory applies to token-level MoE, as attention is typically performed over a batch of tokens. However, we further extend the theory basis (i.e., from *implicit* MoE to *explicit* MoE) to the instance-level MoE and provide theoretical proofs (in Appendix A), by which we reformulate the layer output into an instance-level *explicit* MoE through hierarchical routing and weighted aggregation. Additionally, we conducted experiments to empirically compare token-level and instance-level routing, and the results demonstrate that the latter outperforms the former on MOCO problem (see Appendix E). Therefore, we adopt instance-level routing in our MoE to provide a stronger inductive bias for model learning and optimization. Accordingly, in the remainder of this paper, we adopt the instance-level perspective in PA-MoE.

## 4.2 INTEGRATING PA-MoE INTO THE *Single-Model* PARADIGM

**MoE Module Placement.** As illustrated in Figure 1, the proposed MoE module supports integration into both interaction strategies (a) and (b) in *single-model* paradigm, enabling preference-aware expert selection in both the encoder and decoder in Transformer architectures for each subproblem instance. Specifically, we integrate MoE modules by 1) replacing the feed-forward networks in the encoder; 2) replacing the attention output projection layer of the decoder, which produces the context embeddings used to calculate node selection logits.

The MoE module consists of a set of $m$ experts $\{E_1, E_2, \ldots, E_m\}$. Each expert $E_k$ is instantiated as a self-contained feed-forward neural network that processes input independently and maintains an individual subset of trainable parameters. This design allows experts to specialize in different subspaces of the input distribution. The output of the MoE is obtained by a weighted sum of expert outputs:

$$\text{MoE}(h, \lambda) = \sum_{k=1}^{m} w_k(h, \lambda) \cdot E_k(h), \tag{6}$$

where $w_k(h, \lambda)$ denotes the gating weight generated by the gating network for the expert $k$, based on the representation of input $h$ and preference $\lambda$. The final output is computed by aggregating expert outputs using the gating weights. To ensure generality and computational efficiency, a sparse routing mechanism based on Top-$K$ selection is adopted, where only $K$ experts with the highest gating scores are activated, while the rest are masked out. To encourage a balanced use of experts, we adopt the auxiliary load balancing loss Shazeer et al. (2017) (see Appendix G) to penalize uneven activation frequencies between experts during training.

**Preference-Aware Gating Mechanism.** Gating network determines how input is routed between experts. Although a straightforward gating mechanism routes input solely based on node features, disregarding the semantic variability induced by different preferences, this may result in suboptimal expert allocation across heterogeneous subproblems. Intuitively, the router should be responsible for selecting a sparse subset of experts based on both instance features and preferences. To this end, we propose preference-aware routing strategy. Specifically, the gating network is jointly conditioned on the node embeddings and the preference to modulate the expert selection process. Given a node embedding $h \in \mathbb{R}^d$ and preferences $\lambda \in \mathbb{R}^p$, the gating logits are computed as:

$$r_k(h, \lambda) = h^\top W_k^{(x)} + \lambda^\top W_k^{(\lambda)}, \tag{7}$$

where $W_k^{(x)} \in \mathbb{R}^d$ and $W_k^{(\lambda)} \in \mathbb{R}^p$ are trainable parameters associated with expert $k$. During training, Gaussian noise $\epsilon_k$ is added to encourage exploration and prevent expert collapse:

$$\tilde{r}_k(h, \lambda) = r_k(h, \lambda) + \epsilon_k, \quad \epsilon_k \sim \mathcal{N}(0, \sigma^2). \tag{8}$$

The routing weights are obtained via a softmax over the noisy logits, followed by a Top-$K$ selection to enforce sparse activation:

$$w_k(h, \lambda) = \frac{\exp(\tilde{r}_k(h, \lambda))}{\sum_j \exp(\tilde{r}_j(h, \lambda))}. \tag{9}$$

Notably, the linear preference-aware routing mechanism delivers efficient preference-specific expert allocation with negligible GPU overhead, outperforming more sophisticated gating designs. We empirically validate these advantages in the experimental section.

## 5 EXPERIMENTS

**Problems.** We validate the effectiveness of our method on the bi-objective and tri-objective Traveling Salesman Problem (Bi-TSP and Tri-TSP) Lust & Teghem (2010), bi-objective Capacitated Vehicle Routing Problem (Bi-CVRP) Zajac & Huber (2021) and multi-objective knapsack problem (MOKP) Ishibuchi et al. (2015). The detailed definition of the problem is provided in Appendix B. For the experiments, we use three standard instance sizes for each problem: $n = 20, 50$, and $100$ for MOTSP and MOCVRP, and $n = 50, 100$, and $200$ for MOKP. We also assess the out-of-distribution generalization of the model in larger instances, including Bi-TSP150/200 and three TSPLIB Audet et al. (2021) benchmark instances (KroAB100/150/200).

Table 1: Comparison results on Bi-TSP with 200 random instances.

| Method | 20 Nodes | | | 50 Nodes | | | 100 Nodes | | |
|---|---|---|---|---|---|---|---|---|---|
| | HV ↑ | Gap ↓ | Time | HV ↑ | Gap ↓ | Time | HV ↑ | Gap ↓ | Time |
| WS-LKH | 0.6270 | 0.02% | 10.1m | 0.6415 | -0.02% | 1.8h | 0.7090 | -0.29% | 6.0h |
| MOGLS | 0.6273 | −0.03% | 2.8h | 0.6317 | 1.51% | 7.3h | 0.6854 | 3.04% | 12.4h |
| PPLS/D-C | 0.6256 | 0.24% | 26.3m | 0.6282 | 2.06% | 2.8h | 0.6844 | 3.18% | 11.5h |
| DRL-MOA | 0.6253 | 0.28% | 7s | 0.6338 | 1.18% | 10s | 0.6962 | 1.51% | 19s |
| MDRL | 0.6271 | 0.0% | 6s | 0.6364 | 0.78% | 9s | 0.6969 | 1.41% | 17s |
| EMNH | 0.6271 | 0.0% | 6s | 0.6364 | 0.78% | 9s | 0.6969 | 1.41% | 16s |
| PMOCO | 0.6262 | 0.14% | 6s | 0.6350 | 0.99% | 9s | 0.6969 | 1.41% | 16s |
| **PA-MoE-P** | 0.6280 | −0.14% | 12s | 0.6388 | 0.41% | 31s | 0.7013 | 0.79% | 31s |
| CNH | 0.6269 | 0.03% | 6s | 0.6367 | 0.73% | 9s | 0.6985 | 1.20% | 13s |
| **PA-MoE-C** | 0.6273 | −0.03% | 9s | 0.6375 | 0.61% | 17s | 0.6991 | 1.10% | 30s |
| WE-CA | 0.6270 | 0.02% | 7s | 0.6392 | 0.34% | 10s | 0.7034 | 0.50% | 21s |
| **PA-MoE-W** | 0.6272 | −0.02% | 14s | 0.6400 | 0.22% | 48s | 0.7043 | 0.37% | 31s |
| MDRL-aug | 0.6271 | 0.00% | 33s | 0.6408 | 0.09% | 1.7m | 0.7022 | 0.66% | 14m |
| EMNH-aug | 0.6271 | 0.00% | 33s | 0.6408 | 0.09% | 1.7m | 0.7023 | 0.65% | 14m |
| PMOCO-aug64 | 0.6280 | −0.14% | 48s | 0.6409 | 0.08% | 2.3m | 0.7025 | 0.62% | 9.2m |
| **PA-MoE-P-aug32** | **0.6284** | **-0.21%** | 39s | 0.6414 | 0.00% | 2.2m | 0.7042 | 0.38% | 8.4m |
| **PA-MoE-P-aug64** | **0.6284** | **-0.21%** | 75s | **0.6418** | **-0.06%** | 5.1m | 0.7046 | 0.33% | 16.4m |
| CNH-aug64 | 0.6280 | −0.14% | 53s | 0.6411 | 0.05% | 2.1m | 0.7030 | 0.55% | 9.34m |
| **PA-MoE-C-aug32** | 0.6280 | −0.14% | 50s | 0.6409 | 0.08% | 2.1m | 0.7028 | 0.58% | 9.1m |
| **PA-MoE-C-aug64** | 0.6281 | −0.16% | 1.1m | 0.6413 | 0.01% | 4.3m | 0.7033 | 0.51% | 16.8m |
| WE-CA-aug64 | 0.6271 | 0.00% | 1.2m | 0.6413 | 0.01% | 3.3m | 0.7066 | 0.04% | 16m |
| **PA-MoE-W-aug32** | 0.6271 | 0.00% | 1.1m | 0.6413 | 0.01% | 5.1m | 0.7066 | 0.04% | 17.1m |
| **PA-MoE-W-aug64** | 0.6271 | 0.00% | 1.9m | 0.6414 | 0.00% | 8.5m | **0.7069** | **0.00%** | 37.3m |

Table 2: Comparison results on Tri-TSP with 200 random instances.

| Method | 20 Nodes | | | 50 Nodes | | | 100 Nodes | | |
|---|---|---|---|---|---|---|---|---|---|
| | HV ↑ | Gap ↓ | Time | HV ↑ | Gap ↓ | Time | HV ↑ | Gap ↓ | Time |
| WS-LKH | 0.4712 | 0.00% | 12m | 0.4440 | -0.32% | 1.9h | 0.5076 | -1.13% | 6.6h |
| MOGLS | 0.4627 | 1.80% | 1.5h | 0.4235 | 4.32% | 4.2h | 0.4328 | 13.76% | 13.5h |
| PPLS/D-C | 0.4701 | 0.23% | 1.3h | 0.4301 | 2.82% | 3.7h | 0.4489 | 10.56% | 14.4h |
| DRL-MOA | 0.4694 | 0.38% | 6s | 0.4309 | 2.64% | 11s | 0.4879 | 2.79% | 17s |
| MDRL | 0.4699 | 0.28% | 5s | 0.4317 | 2.46% | 9s | 0.4852 | 3.32% | 19s |
| EMNH | 0.4699 | 0.28% | 5s | 0.4324 | 2.30% | 9s | 0.4866 | 3.05% | 16s |
| PMOCO | 0.4706 | 0.13% | 7s | 0.4346 | 1.81% | 9s | 0.4902 | 2.33% | 17s |
| **PA-MoE-P** | 0.4727 | -0.32% | 12s | 0.4401 | 0.56% | 21s | 0.4965 | 1.07% | 40s |
| CNH | 0.4712 | 0.00% | 6s | 0.4374 | 1.17% | 9s | 0.4926 | 1.85% | 15s |
| **PA-MoE-C** | 0.4715 | -0.06% | 15s | 0.4382 | 0.99% | 29s | 0.4933 | 1.71% | 54s |
| WE-CA | 0.4707 | -0.11% | 5s | 0.4389 | 0.84% | 9s | 0.4975 | 0.88% | 20s |
| **PA-MoE-W** | 0.4708 | -0.09% | 17s | 0.4387 | 0.88% | 25s | 0.4970 | 0.98% | 31s |
| MDRL-aug | 0.4712 | 0.00% | 2.6m | 0.4408 | 0.41% | 25m | 0.4958 | 1.21% | 1.7h |
| EMNH-aug | 0.4712 | 0.00% | 2.6m | 0.4418 | 0.18% | 25m | 0.4973 | 0.92% | 1.7h |
| PMOCO-aug64 | 0.4740 | -0.59% | 42s | 0.4430 | -0.09% | 2.4m | 0.4967 | 1.04% | 9.3m |
| **PA-MoE-P-aug32** | 0.4740 | -0.59% | 33s | 0.4447 | −0.47% | 2.2m | 0.5014 | 0.10% | 9.2m |
| **PA-MoE-P-aug64** | **0.4740** | **-0.59%** | 1.0m | **0.4452** | **-0.58%** | 4.3m | 0.5020 | -0.02% | 17.1m |
| CNH-aug64 | 0.4724 | -0.25% | 42s | 0.4417 | 0.20% | 2.2m | 0.4974 | 0.90% | 9.4m |
| **PA-MoE-C-aug32** | 0.4725 | −0.28% | 35s | 0.4423 | 0.07% | 2.5m | 0.4975 | 0.88% | 8.5m |
| **PA-MoE-C-aug64** | 0.4725 | −0.28% | 1.0m | 0.4428 | -0.04% | 4.5m | 0.4983 | 0.72% | 20.0m |
| WE-CA-aug64 | 0.4712 | 0.00% | 2.2m | 0.4426 | 0.0% | 5.2m | **0.5023** | **-0.08%** | 25.7m |
| **PA-MoE-W-aug32** | 0.4712 | 0.00% | 1.1m | 0.4422 | 0.09% | 5.1m | 0.5014 | 0.10% | 23.3m |
| **PA-MoE-W-aug64** | 0.4712 | 0.00% | 2.9m | 0.4426 | 0.00% | 9.8m | 0.5019 | 0.00% | 39.9m |

**Hyperparameters.** The proposed PA-MoE-P, PA-MoE-C, and PA-MoE-W are based on three representative preference-aware neural MOCO frameworks: PMOCO, CNH, and WE-CA. Most hyperparameters are inherited from their respective baselines, except for the number of experts and the top-$k$ selection size, which are consistently set to 4 and 2, respectively, in all PA-MoE variants. We generate $N$ preferences using the Normal-Boundary Intersection (NBI) method Das & Dennis (1998), with $N = 101$ for bi-objective ($M$=2) and $N = 105$ for tri-objective ($M$=3) problems. The model is trained for 200 epochs, with each epoch consisting of 100,000 randomly sampled instances. We use a batch size of 64 and optimize the network using the Adam optimizer Kingma & Ba (2015) with a learning rate of $10^{-4}$ and a weight decay of $10^{-6}$.

**Baselines.** We compare our method against six learnable and four non-learnable MOCO baselines. (1) Learnable MOCO methods. We compare our method with three representative single-model neural MOCO baselines: **PMOCO** Lin et al. (2022), **CNH** Fan et al. (2024), and **WE-CA** Chen et al. (2025). In addition, we include multi-model baselines, namely **DRL-MOA** Li et al. (2020), and two meta-learning based multi-model approaches, **MDRL** Zhang et al. (2023) and **EMNH** Chen et al. (2023). (2) Non-learnable MOCO methods. **MOGLS** Jaszkiewicz (2002) is a genetic local search MOEA that runs for 4,000 iterations with 100 local search steps per iteration. **PPLS/D-C** Shi et al. (2022) is a specialized MOEA that applies 2-opt heuristics for TSP and CVRP, and a greedy transformation heuristic for KP, with 200 iterations in total. **WS-LKH** and **WS-DP** are weighted-sum (WS) based solvers that scalarize MOTSP and MOKP into single-objective subproblems, which are then solved using LKH heuristic Tinós et al. (2018) and dynamic programming, respectively.

Table 3: Comparison results on Bi-CVRP with 200 random instances.

| Method | 20 Nodes | | | 50 Nodes | | | 100 Nodes | | |
|---|---|---|---|---|---|---|---|---|---|
| | HV ↑ | Gap ↓ | Time | HV ↑ | Gap ↓ | Time | HV ↑ | Gap ↓ | Time |
| MOGLS | 0.4281 | 0.47% | 7.9h | 0.3987 | 2.90% | 17h | 0.3721 | 8.89% | 21h |
| PPLS/D-C | 0.4273 | 0.65% | 1.7h | 0.4012 | 2.29% | 9.3h | 0.3829 | 6.24% | 19h |
| DRL-MOA | 0.4279 | 0.51% | 7s | 0.4075 | 0.75% | 11s | 0.4039 | 1.10% | 24s |
| MDRL | 0.4291 | 0.23% | 8s | 0.4082 | 0.58% | 13s | 0.4056 | 0.69% | 32s |
| EMNH | 0.4299 | 0.05% | 7s | 0.4098 | 0.19% | 13s | 0.4072 | 0.29% | 31s |
| PMOCO | 0.4412 | −2.58% | 6s | 0.4031 | 1.83% | 11s | 0.4025 | 1.44% | 15s |
| **PA-MoE-P** | 0.4414 | −2.63% | 9s | 0.4048 | 1.41% | 18s | 0.4136 | −1.27% | 33s |
| CNH | 0.4437 | −3.16% | 7s | 0.4059 | 1.14% | 11s | 0.4144 | −1.47% | 24s |
| **PA-MoE-C** | 0.4438 | −3.19% | 11s | 0.4062 | 1.07% | 20s | 0.4152 | −1.66% | 37s |
| WE-CA | 0.4290 | 0.26% | 7s | 0.4089 | 0.41% | 14s | 0.4068 | 0.39% | 26s |
| **PA-MoE-W** | 0.4292 | 0.21% | 11s | 0.4095 | 0.27% | 16s | 0.4074 | 0.24% | 30s |
| MDRL-aug | 0.4294 | 0.16% | 11s | 0.4092 | 0.34% | 36s | 0.4072 | 0.29% | 2.8m |
| EMNH-aug | 0.4302 | −0.02% | 11s | 0.4106 | 0.00% | 35s | 0.4079 | 0.12% | 2.8m |
| PMOCO-aug | 0.4421 | −2.79% | 11s | 0.4059 | 1.14% | 23s | 0.4102 | −0.44% | 1.2m |
| **PA-MoE-P-aug4** | 0.4425 | −2.88% | 9s | 0.4064 | 1.02% | 21s | 0.4148 | −1.57% | 59s |
| **PA-MoE-P-aug8** | 0.4427 | −2.92% | 13s | 0.4068 | 0.93% | 33s | 0.4152 | −1.67% | 1.9m |
| CNH-aug8 | **0.4448** | **-3.42%** | 18s | 0.4074 | 0.78% | 43s | 0.4157 | -1.79% | 2.5m |
| **PA-MoE-C-aug4** | 0.4447 | -3.39% | 15s | 0.4074 | 0.78% | 42s | 0.4162 | -1.91% | 2.0m |
| **PA-MoE-C-aug8** | **0.4448** | **-3.42%** | 26s | 0.4078 | 0.68% | 1.0m | **0.4164** | **-1.96%** | 3.6m |
| WE-CA-aug | 0.4300 | 0.02% | 14s | 0.4103 | 0.07% | 49s | 0.4081 | 0.07% | 3.2m |
| **PA-MoE-W-aug4** | 0.4299 | 0.05% | 14s | 0.4104 | 0.05% | 46s | 0.4082 | 0.05% | 3.2m |
| **PA-MoE-W-aug8** | 0.4301 | 0.00% | 23s | **0.4106** | **0.00%** | 1.3m | 0.4084 | 0.00% | 5.7m |

**Metrics.** We evaluate the performance of the methods using the widely adopted hypervolume (HV) indicator Audet et al. (2021) (see Appendix C), where a higher HV indicates a better approximation set. We report the average HV, the HV gap relative to PA-MoE-W-aug, and the total inference time over a fixed set of 200 test instances.

Methods with the "-aug" apply instance augmentation to improve performance (see Appendix D for details). To emphasize the relative performance of neural MOCO solvers, we exclude the strong baseline WS-LKH when marking the best (second-best) results, which are highlighted in bold (underline) if not significantly different at the 1% level according to the Wilcoxon rank-sum test Wilcoxon (1992). The experiments were carried out on a machine with RTX 3090 GPUs and Intel Xeon Silver 4214R CPUs.

## 5.1 MAIN RESULTS

The comparison results between our unified models (PA-MoE-P/C/W) and a diverse set of MOCO baselines is presented in Table 1–Table 3 for MOTSP and MOCVRP, and in Appendix F for MOKP. From these results, we can safely infer that our proposed framework consistently delivers superior performance across different backbone configurations, highlighting its strong adaptability to varying architectural designs.

When instance augmentation is introduced, our models improve further. Especially, even with only half the augmentation size, PA-MoE demonstrates superior performance. For instance, PA-MoE-P-aug32 outperforms PMOCO-aug64 in all benchmark problems. Compared with multi-model methods, PA-MoE maintains its lead in most cases, except on Bi-KP100, where EMNH achieves the lowest gap (−0.15%). PA-MoE establishes new SOTA results across multiple benchmarks, including Bi-TSP20/50 (PA-MoE-P-aug), Bi-CVRP50 (PA-MoE-W-aug), and Bi-CVRP100 (PA-MoE-C-aug). While PA-MoE does not yet surpass WS-LKH on Bi-TSP100 and Tri-TSP100, WS-LKH demands prohibitive runtimes (e.g., 6 hours on Bi-TSP100). In contrast, PA-MoE produces competitive results within minutes, demonstrating a favorable trade-off between efficiency and solution quality.

## 5.2 ABLATION STUDY

**Gating Mechanism.** We investigate five gating strategies in the PA-MoE-P model on Bi-TSP50: (1) Inst-Only, which routes based solely on instance embeddings; (2) Pref-Only, which uses only preferences for expert selection; (3) Concat, which combines instance and preference features via concatenation before gating; (4) Additive (ours), which separately projects and then adds instance and preference representations; and (5) HyperNet, which leverages a lightweight hypernetwork conditioned on preferences to generate part of the gating parameters. As shown in Figure 2 (a), Pref-Only and Inst-Only suffer from limited performance because they rely exclusively on either preference or

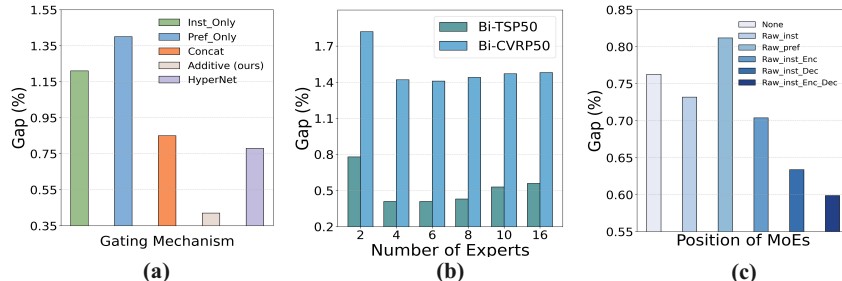

Figure 2: Ablation study results: comparison of gating mechanisms (left), performance under varying numbers of experts (middle), and effects of expert placement (right).

Table 4: Generalization results on 200 instances of larger-size problems.

| Method | 150 Nodes | | | 200 Nodes | | |
|---|---|---|---|---|---|---|
| | HV ↑ | Gap ↓ | Time | HV ↑ | Gap ↓ | Time |
| WS-LKH | 0.7149 | -1.39% | 13h | 0.7490 | -1.39% | 22h |
| MOGLS | 0.6794 | 3.64% | 27h | 0.7181 | 2.79% | 53h |
| PPLS/D-C | 0.6738 | 4.43% | 23h | 0.7087 | 4.06% | 31h |
| DRL-MOA | 0.6919 | 1.87% | 52s | 0.7248 | 1.88% | 1.5m |
| MDRL | 0.6922 | 1.83% | 40s | 0.7251 | 1.84% | 1.4m |
| EMNH | 0.6930 | 1.72% | 40s | 0.7260 | 1.72% | 1.4m |
| PMOCO | 0.6926 | 1.77% | 40s | 0.7261 | 1.71% | 1.1m |
| **PA-MoE-P** | 0.6983 | 0.96% | 1.4m | 0.7323 | 0.87% | 1.7m |
| MDRL-aug | 0.6976 | 1.06% | 47m | 0.7299 | 1.19% | 1.6h |
| EMNH-aug | 0.6983 | 0.96% | 47m | 0.7307 | 1.08% | 1.6h |
| PMOCO-aug64 | 0.6982 | 0.98% | 25.4m | 0.7311 | 1.03% | 49.9m |
| **PA-MoE-P-aug32** | 0.7013 | 0.54% | 23.2m | 0.7350 | 0.50% | 46.4m |
| **PA-MoE-P-aug64** | 0.7018 | 0.47% | 45.0m | 0.7355 | 0.43% | 1.5h |
| CNH | 0.6950 | 1.43% | 1.1m | 0.7295 | 1.24% | 1.2m |
| **PA-MoE-C** | 0.6953 | 1.39% | 1.3m | 0.7297 | 1.22% | 2.0m |
| CNH-aug64 | 0.6997 | 0.77% | 47.9m | 0.7339 | 0.65% | 1.3h |
| **PA-MoE-C-aug32** | 0.6992 | 0.84% | 38.7m | 0.7335 | 0.70% | 1.2h |
| **PA-MoE-C-aug64** | 0.6998 | 0.75% | 1.2h | 0.7340 | 0.64% | 1.9h |
| WE-CA | 0.7008 | 0.61% | 1.2m | 0.7346 | 0.56% | 3.6m |
| **PA-MoE-W** | 0.7019 | 0.45% | 1.8m | 0.7360 | 0.36% | 3.9m |
| WE-CA-aug64 | 0.7044 | 0.10% | 53.6m | 0.7381 | 0.08% | 1.7h |
| **PA-MoE-W-aug32** | 0.7047 | 0.06% | 45.2m | 0.7385 | 0.03% | 1.4h |
| **PA-MoE-W-aug64** | **0.7051** | **0.00%** | 1.1h | **0.7387** | **0.00%** | 1.9h |

instance features. By conditioning the gating mechanism on a single source of information, these models struggle to capture nuanced subproblem-specific features and fail to achieve flexible expert allocation. While modest gains are observed with the other fusion strategies, the Additive variant proves most effective.

**Number of Experts.** Figure 2 (b) reveals that PA-MoE-P benefits from increased expert capacity up to 8, beyond which performance saturates or degrades. These results indicate that simply increasing the number of experts does not consistently improve performance. Similar to challenges observed in single-objective tasks Zhou et al. (2024), overly large expert pools may suffer from insufficient training of parameters and degraded generalization due to under-utilization and increased optimization difficulty.

**Position of MoE.** We evaluate the impact of inserting MoE modules at different positions in the transformer architecture with PMOCO backbone, on the Bi-TSP50 dataset. As shown in Figure 2 (c), we first apply MoE to the linear projections of instance features (Raw-Inst) and preferences (Raw-Pref) separately. We observe that only applying MoE to Raw-Inst results in a performance improvement. Hence, we extend Raw-Inst by incorporating MoE into the encoder's feed-forward layers (Raw-Inst-Enc) and the decoder's attention output projection layer (Raw-Inst-Dec), each yielding additional gains. By combining both encoder-side and decoder-side enhancements, the resulting configuration (Raw-Inst-Enc-Dec) achieves the best performance and is therefore adopted throughout the paper to fully exploit its potential.

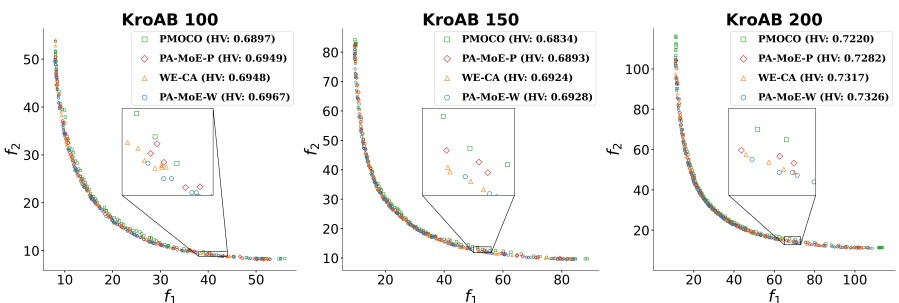

Figure 3: Pareto fronts of benchmark instances, KroAB100/150/200 (left/middle/right).

## 5.3 GENERALIZATION STUDY

Table 5: Comparison results on benchmark instances.

| Method | KroAB100 | | | KroAB150 | | | KroAB200 | | |
|---|---|---|---|---|---|---|---|---|---|
| | HV↑ | Gap↓ | Time↓ | HV↑ | Gap↓ | Time↓ | HV↑ | Gap↓ | Time↓ |
| WS-LKH | 0.7022 | -0.40% | 2.3m | 0.7017 | -0.73% | 4.0m | 0.7430 | -1.05% | 5.6m |
| MOGLS | 0.6819 | 2.50% | 49m | 0.6651 | 4.52% | 1.2h | 0.7045 | 4.19% | 1.5h |
| PPLS/D-C | 0.6785 | 2.99% | 37m | 0.6672 | 4.22% | 49m | 0.7193 | 2.17% | 3.1h |
| DRL-MOA | 0.6904 | 1.29% | 10s | 0.6793 | 2.48% | 18s | 0.7185 | 2.28% | 25s |
| MDRL | 0.6881 | 1.62% | 10s | 0.6831 | 1.94% | 17s | 0.7209 | 1.96% | 23s |
| EMNH | 0.6900 | 1.34% | 9s | 0.6832 | 1.92% | 16s | 0.7217 | 1.85% | 23s |
| PMOCO | 0.6897 | 1.39% | 9s | 0.6834 | 1.89% | 17s | 0.7220 | 1.81% | 23s |
| **PA-MoE-P** | 0.6949 | 0.64% | 11s | 0.6893 | 1.05% | 25s | 0.7282 | 0.96% | 38s |
| CNH | 0.6913 | 1.16% | 13s | 0.6844 | 1.75% | 22s | 0.7241 | 1.52% | 37s |
| **PA-MoE-C** | 0.6924 | 1.00% | 24s | 0.6857 | 1.56% | 34s | 0.7246 | 1.46% | 43s |
| WE-CA | 0.6948 | 0.66% | 12s | 0.6924 | 0.60% | 19s | 0.7317 | 0.49% | 32s |
| **PA-MoE-W** | 0.6967 | 0.39% | 18s | 0.6928 | 0.55% | 23s | 0.7326 | 0.37% | 42s |
| MDRL-Aug | 0.6950 | 0.63% | 13s | 0.6890 | 1.09% | 19s | 0.7261 | 1.25% | 28s |
| EMNH-Aug | 0.6958 | 0.51% | 12s | 0.6892 | 1.06% | 18s | 0.7270 | 1.13% | 27s |
| PMOCO-Aug | 0.6958 | 0.51% | 10s | 0.6902 | 0.92% | 21s | 0.7272 | 1.10% | 45s |
| **PA-MoE-P-Aug** | 0.6985 | 0.13% | 17s | 0.6937 | 0.42% | 28s | 0.7317 | 0.49% | 53s |
| CNH-Aug | 0.6965 | 0.41% | 15s | 0.6911 | 0.79% | 27s | 0.7238 | 1.56% | 48s |
| **PA-MoE-C-Aug** | 0.6973 | 0.30% | 19s | 0.6913 | 0.76% | 32s | 0.7301 | 0.71% | 52s |
| WE-CA-aug | 0.6990 | 0.06% | 22s | 0.6957 | 0.13% | 23s | 0.7349 | 0.54% | 42s |
| **PA-MoE-W-Aug** | **0.6994** | **0.00%** | 28s | **0.6966** | **0.00%** | 29s | **0.7353** | **0.00%** | 55s |

We evaluated the generalization performance of models trained on Bi-TSP100 by testing them on 200 randomly generated instances of larger Bi-TSP150 and Bi-TSP200, as shown in Table 4. Furthermore, we assess performance on three TSPLIB benchmarks, KroAB100, KroAB150, and KroAB200, reported in Table 5. The results show that PA-MoE-P and PA-MoE-W outperform state-of-the-art neural baselines on Bi-TSP150 and Bi-TSP200 benchmark sets. The corresponding Pareto fronts are also visualized in Figure 3. The magnified regions of the Pareto fronts reveal that PA-MoE yields clearly dominant solutions over those produced by baseline, validating the superiority of our approach in balancing convergence and diversity. These results indicate that the PA-MoE architecture is generalizable and corroborate our hypothesis about its robustness.

## 6 CONCLUSION

In this paper, we provide a novel perspective on the single-model paradigm for MOCO by establishing its connection to an implicit Mixture-of-Experts architecture. Subsequently, we propose a generic PA-MoE framework that employs sparsely activated expert modules and an innovative lightweight gating mechanism, thereby explicitly enhancing the capacity of a single model to address diverse subproblems. Extensive results demonstrate PA-MoE's effectiveness and favored cross-size generalization capability. We believe there remains ample room to uncover deeper connections between MOCO and MoE, which could inform the design of more effective neural solvers. For example, incremental expert generation at the subproblem level under shifting preferences, and the design of more intricate expert architectures, are promising directions.

## REPRODUCIBILITY STATEMENT

We are committed to ensuring the reproducibility of our work. To this end, we provide the following: (1) **Code availability:** The complete source code and detailed instructions for reproducing our experiments are included in the supplementary material, which can be accessed and downloaded by reviewers and readers. (2) **Theoretical results:** All assumptions are clearly stated, and complete proofs of our main theorems are presented in Appendices A.1 and A.2.

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

# A  CONSTRUCTIVE DERIVATION FROM TOKEN-LEVEL IMPLICIT MOE TO INSTANCE-LEVEL EXPLICIT MOE

Let $H = [h_1, \ldots, h_N]$ be the hidden states of a layer (e.g., an attention block) for an instance $X$, and $\lambda$ a preference vector. An *implicit* token-level mixture has the per-token output

$$y_i = \sum_{k=1}^{m} w_{ik}(h_i, \lambda) \, E_k(h_i), \qquad w_{ik} \geq 0, \sum_{k=1}^{m} w_{ik} = 1, \tag{10}$$

where $E_k$ are shared (token-agnostic) expert functions and $w_{ik}$ are token-level gates.

## A.1  WEIGHTED AGGREGATION IDENTITY

**Lemma 1** (Weighted aggregation identity). *Let $y_i$ be defined as in equation 10. For any aggregation operator that is a linear mean over tokens (e.g., $\frac{1}{N} \sum_i$), we have the exact identity*

$$F_{\text{inst}}(X, \lambda) := \frac{1}{N} \sum_{i=1}^{N} y_i = \sum_{k=1}^{m} \underbrace{\left( \frac{1}{N} \sum_{i=1}^{N} w_{ik} \right)}_{\pi_k(X, \lambda)} \underbrace{\left( \frac{\sum_{i=1}^{N} w_{ik} E_k(h_i)}{\sum_{i=1}^{N} w_{ik}} \right)}_{\widehat{E}_k(X, \lambda)}. \tag{11}$$

*Proof.* Expand $\frac{1}{N} \sum_i y_i$ using equation 10:

$$\frac{1}{N} \sum_i y_i = \frac{1}{N} \sum_i \sum_k w_{ik} E_k(h_i) = \sum_k \left( \frac{1}{N} \sum_i w_{ik} E_k(h_i) \right).$$

For each $k$, multiply and divide by $\sum_i w_{ik}$ (non-negative; when zero, define the term as 0 by continuity), yielding

$$\sum_k \left( \frac{1}{N} \sum_i w_{ik} \right) \left( \frac{\sum_i w_{ik} E_k(h_i)}{\sum_i w_{ik}} \right) = \sum_k \pi_k \widehat{E}_k.$$

$\square$

## A.2  HIERARCHICAL ROUTING AND INSTANCE-LEVEL MOE

Define an instance-/preference-level context $c$ by pooling token states as:

$$c(X, \lambda) := \text{Pool}\big(H(X, \lambda)\big) \in \mathbb{R}^{d_c} \quad \text{(e.g., mean/max pooling or learned attention pooling).} \tag{12}$$

Then, introduce a *hierarchical* router that factors instance- and token-level gates as:

$$\beta_k(c, \lambda) := \text{softmax}_k\big(u_k^\top \phi([c; \lambda])\big), \tag{13}$$

$$\alpha_{ik}(h_i, \lambda) := \text{softmax}_k\big(a_k^\top \psi([h_i; \lambda])\big), \tag{14}$$

$$\tilde{w}_{ik}(h_i, c, \lambda) \propto \beta_k(c, \lambda) \, \alpha_{ik}(h_i, \lambda), \qquad w_{ik} = \frac{\tilde{w}_{ik}}{\sum_{k'} \tilde{w}_{ik'}}. \tag{15}$$

**Proposition 1** (Constructive instance-level MoE form). *Let $y_i$ be as in equation 10 with gates defined by equation 15. Then the instance-level aggregation $F_{\text{inst}}$ in equation 11 equals*

$$F_{\text{inst}}(X, \lambda) = \sum_{k=1}^{m} \pi_k(X, \lambda) \, \widehat{E}_k(X, \lambda), \quad \pi_k := \frac{1}{N} \sum_i w_{ik}, \quad \widehat{E}_k := \frac{\sum_i w_{ik} E_k(h_i)}{\sum_i w_{ik}}, \tag{16}$$

*which is an* instance-level explicit MoE *with expert responses $\widehat{E}_k$ and gates $\pi_k$.*

If one prefers the canonical structure $\sum_k \beta_k(c, \lambda) \, \widetilde{E}_k(X)$, set:

$$\widetilde{E}_k(X) := \widehat{E}_k(X, \lambda) \quad \text{and} \quad \beta_k(c, \lambda) \propto \pi_k(X, \lambda) \text{ (followed by normalization),} \tag{17}$$

then $F_{\text{inst}}(X, \lambda) = \sum_k \beta_k(c, \lambda) \, \widetilde{E}_k(X)$ holds exactly by construction.

### A.3 Sparsity and capacity control at the instance level

The instance-level gates $\pi_k$ or $\beta_k$ expose explicit control: (i) *Top-K sparsity*: keep the $K$ largest entries and renormalize; (ii) *Load balancing*: add a standard MoE auxiliary loss on average usage,

$$\mathcal{L}_{\text{load}} = m \sum_{k=1}^{m} \bar{p}_k \log \bar{p}_k, \qquad \bar{p}_k := \mathbb{E}_{X,\lambda}\left[\frac{1}{N}\sum_{i=1}^{N} w_{ik}(h_i, c, \lambda)\right]. \tag{18}$$

This operates *at the instance level*, which dense token-indexed routing in standard attention does not directly expose.

### A.4 Expressive Power of Explicit MoE: Approximation and Extension

**Proposition 2** (Approximation and extension). *Assume experts $\{E_k\}_{k=1}^{m}$ are MLPs with a universal approximation property on compact sets, and the router in equation 15 is implemented by MLPs $\phi, \psi$. Then for any token-mixing attention layer (with value projection linear in $H$), the mapping $H \mapsto \frac{1}{N}\sum_i y_i$ can be approximated arbitrarily well by the instance-level explicit MoE in equation 16. Moreover, by choosing non-linear experts, the* explicit *MoE can represent mappings beyond linear token mixing under comparable depth/width.*

*Proof.* (i) *Approximation*: Fix $k$ and approximate $E_k(h)$ to match the target value-map behaviour; let the router mimic attention weights by fitting $w_{ik}$ via equation 15. The weighted identity equation 11 then recovers the aggregated mapping. Universal approximation of MLPs gives the arbitrarily small error. (ii) *Extension*: Non-linear experts allow composition of non-linear token responses aggregated at instance level, which exceeds the linear subspace spanned by token-mixing with a fixed linear $V$; a small counterexample is obtained by requiring expert-specific non-linearities that cannot be reduced to a single linear value map without increasing depth/width. $\qquad\square$

Prop. 2 concerns the *layer-level* representational view after aggregation; it is *not* a claim of equivalence between full model classes.

# B MULTI-OBJECTIVE COMBINATORIAL OPTIMIZATION PROBLEMS

Multi-objective combinatorial optimization problems extend classical optimization problems by incorporating multiple objectives. This section explores three key problems: the Multi-Objective Traveling Salesman Problem (MOTSP), the Multi-Objective Capacitated Vehicle Routing Problem (MOCVRP), and the Multi-Objective Knapsack Problem (MOKP), each involving the optimization of competing objectives under specific constraints.

### B.1 Multi-objective traveling salesman problem (MOTSP)

MOTSP is an extension of the classic single-objective Traveling Salesman Problem (TSP). In MOTSP, $M$ objectives are considered, with each objective represented by a distinct set of node coordinates. The aim is to find a tour $\pi$, which is a cyclic permutation of the nodes, that simultaneously minimizes the costs across all objectives:

$$\min L(\pi|s) = \min(L_1(\pi|s), L_2(\pi|s), \ldots, L_M(\pi|s)), \tag{19}$$

where $L_i(\pi|s)$ denotes the cost for the $i$-th objective and is calculated as:

$$L_i(\pi|s) = c_i(\pi(n), \pi(1)) + \sum_{j=1}^{n-1} c_i(\pi(j), \pi(j+1)). \tag{20}$$

Here, $c_i(j, k)$ represents the cost of moving from node $j$ to node $k$ under the $i$-th objective. The solution to MOTSP often involves trade-offs as it requires minimizing all objective functions simultaneously.

### B.2 MULTI-OBJECTIVE CAPACITATED VEHICLE ROUTING PROBLEM (MOCVRP)

MOCVRP aims to optimize two objectives simultaneously: minimizing the total length of the route, which is the sum of distances traveled by all vehicles, and minimizing the makespan, defined as the length of the longest route. This problem involves a depot node and multiple customer nodes, each with a specific demand $q_i$. A fleet of vehicles, each with a fixed capacity $D$, starts and ends its routes at the depot, ensuring that the total demand on any route satisfies the constraint $\sum q_i \leq D$.

The total route length can be mathematically formulated as

$$\min f_1(\pi) = \sum_{k=1}^{K} \sum_{i=1}^{n_k} d_{\pi_k(i), \pi_k(i+1)}, \tag{21}$$

where $K$ denotes the number of vehicles, $n_k$ is the number of customer nodes in the $k$-th route, and $d_{\pi_k(i), \pi_k(i+1)}$ is the distance between consecutive nodes in the route. The makespan, representing the longest route among all vehicles, is expressed as

$$\min f_2(\pi) = \max_{k \in \{1, \ldots, K\}} \sum_{i=1}^{n_k} d_{\pi_k(i), \pi_k(i+1)}. \tag{22}$$

In addition, the solution must satisfy two key constraints. Each customer must be visited exactly once, and all routes must start and end at the depot. This problem models real-world scenarios where optimizing operational efficiency and resource utilization is critical in multi-vehicle delivery systems.

### B.3 MULTI-OBJECTIVE KNAPSACK PROBLEM (MOKP)

The Knapsack Problem (KP) is a classic problem in combinatorial optimization, and MOKP is an extension of KP, involving $m$ objectives and $n$ items. The goal of this problem is to maximize the values of multiple objective functions:

$$f(x) = \max(f_1(x), f_2(x), \ldots, f_m(x)), \tag{23}$$

where each objective function is defined as

$$f_i(x) = \sum_{j=1}^{n} v_{ij} x_j. \tag{24}$$

The constraints are given by

$$\sum_{j=1}^{n} w_j x_j \leq W, \quad \text{with } x_j \in \{0, 1\}. \tag{25}$$

Each item has a weight $w_j$ and $m$ different values $v_{ij}$, where $i = 1, 2, ..., m$. The knapsack has a maximum weight capacity $W$, and the objective is to select a set of items such that their total weight does not exceed the capacity $W$, while maximizing the sum of values for each objective.

## C HYPERVOLUME INDICATOR

The hypervolume (HV) indicator is one of the most widely adopted metrics for evaluating the quality of solution sets in multi-objective combinatorial optimization (MOCO), as it comprehensively measures both the convergence and diversity of the obtained Pareto front without requiring access to the ground truth. Given a reference point $r \in \mathbb{R}^M$ and a Pareto front $\mathcal{F}$, the HV indicator is defined as:

$$\text{HV}_r(\mathcal{F}) = \mu \left( \bigcup_{f(x) \in \mathcal{F}} [f(x), r] \right), \tag{26}$$

where $\mu$ denotes the Lebesgue measure, and $[f(x), r]$ denotes the hyper-rectangle spanned between a solution $f(x)$ and the reference point $r$. For example, in the bi-objective case, the HV value corresponds to the area of the union of rectangles extending from each solution to the reference point in the objective space.

Since HV values are sensitive to the scale of the objective space, we employ a normalized version defined as:

$$\text{HV}'_r(\mathcal{F}) = \frac{\text{HV}_r(\mathcal{F})}{\prod_{i=1}^{M} |r_i - z_i|}, \tag{27}$$

where $z$ is an ideal point satisfying $z_i < \min\{f_i(x) \mid f(x) \in \mathcal{F}\}$ for all $i \in \{1, \dots, M\}$ in the case of minimization problems. This normalization ensures fair comparison across different problem instances and objective scales.

From a theoretical perspective, optimizing the HV indicator over a fixed-size population of $\mu$ solutions corresponds to maximizing a single-objective quality indicator. The resulting set of $\mu$ solutions is referred to as an *optimal $\mu$-distribution*, which introduces an implicit search bias depending on the indicator and the choice of the reference point.

The reference point $r$ plays a crucial role in determining which regions of the objective space are favored during optimization. A poorly chosen reference point can lead to suboptimal or biased distributions, especially with respect to extreme solutions on the Pareto front. In this work, we adopt a consistent reference point across all compared methods, and the specific values used are summarized in Table 6.

Table 6: Reference points and ideal points.

| Problem | Size | r | z |
|---------|------|-----|-----|
| Bi-TSP | 20 | (20, 20) | (0, 0) |
| | 50 | (35, 35) | (0, 0) |
| | 100 | (65, 65) | (0, 0) |
| | 150 | (85, 85) | (0, 0) |
| | 200 | (115, 115) | (0, 0) |
| Bi-CVRP | 20 | (30, 4) | (0, 0) |
| | 50 | (45, 4) | (0, 0) |
| | 100 | (80, 4) | (0, 0) |
| Bi-KP | 50 | (5, 5) | (30, 30) |
| | 100 | (20, 20) | (50, 50) |
| | 200 | (30, 30) | (75, 75) |
| Tri-TSP | 20 | (20, 20, 20) | (0, 0, 0) |
| | 50 | (35, 35, 35) | (0, 0, 0) |
| | 100 | (65, 65, 65) | (0, 0, 0) |

## D  INSTANCE AUGMENTATION

To further improve the performance during inference, we adopt an instance-level augmentation strategy Kwon et al. (2020) Lin et al. (2022) that generates multiple geometric transformations of the same input.

In single-objective combinatorial optimization, Euclidean problem instances such as TSP and CVRP are typically defined with node coordinates sampled from the unit square $[0, 1]^2$. Given the invariance of pairwise Euclidean distances under certain geometric transformations, the underlying problem structure remains unchanged when the coordinates are transformed through operations such as reflection, rotation, or axis swapping. A commonly used set of eight symmetry-preserving transformations is defined over the unit square. Given any coordinate $(x, y)$, the transformed coordinates $(x', y')$ are selected from the set $\{(x, y), (y, x), (x, 1-y), (1-x, y), (1-x, 1-y), (1-y, x), (y, 1-x), (1-y, 1-x)\}$. These transformations preserve the essential structure of the problem, while neural models may respond differently to varied input representations. As a result, the generated

solutions remain feasible for the original instance and, in some cases, can even outperform those obtained from the original input, serving as better approximations to the optimal solution.

In this paper, preference vectors are generated using the Normal-Boundary Intersection (NBI) method Das & Dennis (1998). For each preference, input augmentations are applied while keeping the scalarization fixed. The augmented instances are decoded independently, and their solutions are mapped back to the original coordinate space and evaluated under the same objective formulation. When each objective is associated with a separate coordinate space, as in Bi-TSP or Tri-TSP, geometric transformations should be applied independently to each objective-specific embedding. Since each coordinate set can undergo eight distinct symmetric transformations, a problem with $M$ objectives yields $8^M$ transformed variants. In particular, Bi-TSP results in $8^2 = 64$ transformations, and Tri-TSP results in $8^3 = 512$. For Bi-CVRP, where each node has only one coordinate set, there are eight possible transformations per instance. Consequently, the best solution is selected by evaluating the objective values across both the original and all transformed instances.

# E   TOKEN VS. INSTANCE-LEVEL ROUTING (GATING) ABLATIONS

Table 7: Token-level vs. instance-level routing in PA-MoE on Bi-TSP50 and Bi-CVRP50.

| Backbone | Routing | Bi-TSP50 | | Bi-CVRP50 | |
|---|---|---|---|---|---|
| | | HV $\uparrow$ | Time (s) $\downarrow$ | HV $\uparrow$ | Time (s) $\downarrow$ |
| PMOCO | Token-level | 0.6379 | 31s | 0.4041 | 18s |
| | Instance-level (PA-MoE-P) | 0.6388 | 31s | 0.4048 | 18s |
| WE-CA | Token-level | 0.6393 | 45s | 0.4093 | 15s |
| | Instance-level (PA-MoE-W) | 0.6400 | 48s | 0.4095 | 16s |

We compare **Token-level** routing ($w_{ik} = \alpha_{ik}(h_i, \lambda)$) and **Instance-level (hierarchical)** routing ($w_{ik} \propto \beta_k(c, \lambda)\, \alpha_{ik}(h_i, \lambda)$, $c = \text{Pool}(H)$). Here, $H = [h_1, \ldots, h_N]$ are token states, $\text{Pool}(\cdot)$ is an instance/pref. pooling (e.g., mean/attn), $\alpha_{ik}$ is a token-level gating score, $\beta_k$ is an instance-level gating score, and $\propto$ indicates subsequent normalization (and Top-$K$ selection) to obtain a valid distribution over experts. As shown in Table 7, instance-level routing (PA-MoE) consistently yields slightly higher HV on both Bi-TSP50 and Bi-CVRP50, with negligible overhead in runtime (e.g., 31s vs 31s). This supports the view that instance-aware capacity allocation is beneficial even under the same computational cost.

# F   MOKP RESULTS

Table 8: Comparison results on Bi-KP with 200 random instances.

| Method | 50 Nodes | | | 100 Nodes | | | 200 Nodes | | |
|---|---|---|---|---|---|---|---|---|---|
| | HV $\uparrow$ | Gap $\downarrow$ | Time | HV $\uparrow$ | Gap $\downarrow$ | Time | HV $\uparrow$ | Gap $\downarrow$ | Time |
| WS-DP | 0.3561 | -0.11% | 22m | 0.4532 | -0.09% | 2.0h | 0.3601 | 0.00% | 5.8h |
| MOGLS | 0.3532 | 0.70% | 5.1h | 0.4502 | 0.57% | 7.3h | 0.3517 | 2.33% | 13h |
| PPLS/D-C | 0.3517 | 1.12% | 23m | 0.4398 | 2.87% | 48m | 0.3527 | 2.05% | 1.2h |
| DRL-MOA | 0.3557 | 0.00% | 9s | 0.4531 | -0.07% | 22s | 0.3601 | 0.00% | 1.1m |
| MDRL | 0.3530 | 0.76% | 6s | 0.4532 | -0.09% | 21s | 0.3601 | 0.00% | 1.2m |
| EMNH | **0.3561** | **-0.11%** | 6s | **0.4535** | **-0.15%** | 21s | **0.3603** | **-0.06%** | 1.2m |
| PMOCO | 0.3521 | 1.01% | 7s | 0.4445 | 1.83% | 16s | 0.3587 | 0.39% | 1.2m |
| **PA-MoE-P** | 0.3547 | 0.28% | 9s | 0.4450 | 1.72% | 22s | 0.3590 | 0.30% | 1.4m |
| WE | 0.3554 | 0.08% | 9s | 0.4527 | 0.02% | 19s | 0.3601 | 0.00% | 1.1m |
| **PA-MoE-W** | 0.3557 | 0.00% | 13s | 0.4528 | 0.00% | 30s | 0.3601 | 0.00% | 1.6m |

As shown in Table 8, the PA-MoE framework exhibits consistent advantages on the Bi-KP problem across varying sizes (50/100/200 items). Both variants (PA-MoE-P/W) outperform PMOCO on BiKP50/100, and PA-MoE-P further maintains its edge on BiKP200 while PA-MoE-W matches the baseline. This demonstrates the strong scalability and competitive generalization of PA-MoE.

## G    DETAILS OF AUXILIAR LOAD BALANCING LOSS

---

**Algorithm 1 Training algorithm**

---

**Input:** weight distribution $\Lambda$, instance distribution $\mathcal{S}_n$ on problem size $n$, number of training steps $E$, batch size $B$, optimizer ADAM
**Output:** Learned model parameters $\theta$
 1: Initialize the model parameters $\theta$
 2: **for** $e = 1$ to $E$ **do**
 3:     Sample $s_i \sim$ SampleInstance($\mathcal{S}_n$)    $\forall i \in \{1, \cdots, B\}$
 4:     Sample $\lambda \sim$ SampleWeight($\Lambda$)
 5:     **for** $i = 1$ to $B$ **do**
 6:         Sample $\pi_i \sim$ SampleSolution($P(\pi|\lambda, s_i)$)
 7:         $g_i \leftarrow g(\pi_i|\lambda, s_i)$
 8:         $\nabla \mathcal{J}^{(i)}(\theta) \leftarrow (g_i - b_i) \nabla_\theta \log P(\pi_i|\lambda, s_i)$
 9:         Compute load balancing loss $\mathcal{L}^{(i)}_{\text{balance}}$ based on CV of importance and load
10:     **end for**
11:     $\nabla \mathcal{J}(\theta) \leftarrow \frac{1}{B} \sum_{i=1}^{B} \nabla \mathcal{J}^{(i)}(\theta)$
12:     $\mathcal{L}_{\text{balance}} \leftarrow \sum_{i=1}^{B} \mathcal{L}^{(i)}_{\text{balance}}$
13:     $\theta \leftarrow$ ADAM($\theta, \nabla \mathcal{J}(\theta) + \nabla \mathcal{L}_{\text{balance}}$)
14: **end for**

---

The proposed PA-MoE adopts instance-level gating, wherein each input instance computes a gating score vector to select its Top-K preferred experts. This routing strategy enables greater flexibility, as different instances may activate distinct sets of experts. However, this flexibility comes at the cost of increased difficulty in balancing expert load, often leading to expert collapse, where only a small subset of experts are frequently utilized while others remain idle. Following Shazeer et al. (2017), we incorporate input-dependent noise into the gating logits to facilitate stochastic expert selection and optimize expert assignment via load-balancing loss for each subproblem in MOCO.

**Load-Balancing Loss**    To learn balanced expert usage, an auxiliary loss is introduced based on the coefficient of variation (CV) of both expert importance and estimated load. This regularization encourages uniform expert activation and enables effective gradient-based optimization of the gating parameters. Specifically, let $G \in \mathbb{R}^m$ denote the gating vector produced by the top-$k$ mechanism for an input instance, where $m$ is the number of experts. The importance of expert $j$ is defined as:

$$\text{importance}_j = G[j], \tag{28}$$

representing the gating weight received by expert $j$ from the current instance. We approximate the expert load, which reflects the expected number of instances routed to each expert, using a smooth and differentiable formulation. This is achieved by modeling the routing probability from instance $s$ to expert $j$ using the cumulative distribution function (CDF) of a standard normal distribution:

$$\Pr(s \rightarrow j) = \Phi\left(\frac{H[j] - T_k}{\sigma_j}\right), \tag{29}$$

where $H[j]$ is the gating logit for expert $j$ of the instance; $T_k$ is the $k$-th largest gating score among all experts excluding $j$; and $\sigma_j$ is the input-dependent noise scale computed as:

$$\sigma_j = \text{Softplus}((hW_{\text{noise}})_j) + \epsilon_{\text{noise}}, \tag{30}$$

with $h$ being the instance embedding. The soft load is then computed across a mini-batch as:

$$\tilde{\text{load}}_j = \sum_{i=1}^{B} \Pr(s_i \rightarrow j). \tag{31}$$

To quantify expert imbalance, the coefficient of variation (CV) is adopted as a normalized metric to measure dispersion across expert usage. Specifically, the final load-balancing loss is defined as:

$$\mathcal{L}_{\text{balance}} = \lambda_{\text{bal}} \cdot \left(\text{CV}^2(\text{importance}) + \text{CV}^2(\tilde{\text{load}})\right), \tag{32}$$

where $\lambda_{\text{bal}}$ is a weighting coefficient set to 0.01 in our experiments. This loss encourages the gating network to maintain both balanced gating weights and equitable expert selection frequencies. The

auxiliary load-balancing loss $\mathcal{L}_{\text{balance}}$ is combined with the scalarized objective loss of each sub-problem to form the overall training objective. These losses are jointly optimized via gradient-based training, enabling each subproblem to participate in backpropagation and contribute to the global parameter updates. The complete training procedure is presented in Algorithm 1.

