# OpenReview forum: "Preference-Aware Mixture-of-Experts for Multi-Objective Combinatorial Optimization"
_ICLR.cc/2026/Conference — ICLR 2026 Conference Withdrawn Submission_

### Official Review · Reviewer_AkxG · 2025-10-26

**Soundness:** 1
**Presentation:** 2
**Contribution:** 1
**Rating:** 2
**Confidence:** 4

**Summary:**

This paper proposes a mixture-of-experts (MoE) extension to transformer-based deep reinforcement learning methods for multi-objective combinatorial optimization. The authors draw a conceptual connection between transformers and MoE, viewing attention as an implicit dense mixture, and then introduce an explicit preference-aware MoE with sparse gating conditioned on instance and preference embeddings. The method is evaluated on several MOCO benchmarks (e.g., multi-objective TSP, CVRP, and knapsack) and reports modest improvements in hypervolume metrics over existing transformer-based baselines. However, the paper overstates its theoretical contribution, the empirical performance improvement is very limited, and the experiment setting lacks rigour, making the reported results unreliable.

**Strengths:**

1. The topic of improving transformer-based DRL for multi-objective combinatorial optimization is timely and relevant to the research community.
2. The proposed MoE architecture is conceptually simple and can be easily integrated into existing transformer backbones.
3. The paper provides evaluations on several benchmark problems (TSP, CVRP, and knapsack), showing that the method can be applied across multiple MOCO settings.

**Weaknesses:**

1. The claimed theoretical analysis in Section 4.1 (Eq. 3–4) is superficial. The comparison between preference-conditioned attention and MoE merely restates that attention is a weighted sum over tokens, without offering new insight.
2. In section 4.2, the proposed PA-MoE method simply integrates an MoE module into the single-model DRL framework, without explaining how different experts correspond to distinct regions of the preference space. In other words, the design is weakly related to multi-objective learning and does not effectively leverage the preference information.
3. In the experiment section, the reported improvements are modest, and the paper provides only single-run results without reporting variance or confidence intervals. As a result, the observed gains may fall within random noise or implementation details and augmentation tricks (as suggested in Appendix D), rather than representing consistent algorithmic improvements. The authors also fail to report several important experimental settings, which makes it difficult to assess the fairness of the results (see questions below).

**Questions:**

1. The proposed PA-MoE adds preference information to the gating function. Is there any mechanism or evidence showing that different experts specialize in different regions of the preference space?
2. How does the proposed design specifically address challenges unique to multi-objective learning, rather than simply adding MoE to a transformer backbone?
3. Did the authors run multiple seeds, and if so, what are the mean and standard deviation across runs?
4. In most of the results reported in Tables 1,2,3, and 4, the HV improvement of PA-MoE is less than 1%, especially when with "-aug", while the inference time is obviously longer. Could the authors justify why these are considered meaningful?
5. The paper reports only HV as the evaluation metric. However, in single-model multi-objective reinforcement learning, testing more preference vectors potentially yields more Pareto-optimal policies and thus a higher HV (with longer inference time). Did the authors ensure that the number of tested preferences is the same across all baselines? The authors also compare with multi-model baselines (e.g., MDRL, EMNH). However, in these methods, the number of models or policies trained can strongly influence the final performance, and this information is not reported.

---

### Official Review · Reviewer_PWEN · 2025-10-28

**Soundness:** 2
**Presentation:** 2
**Contribution:** 2
**Rating:** 2
**Confidence:** 4

**Summary:**

This paper proposes a preference-aware mixture-of-experts framework for multi-objective combinatorial optimization problems. The authors first draw an analogy between the attention layer in Transformers and the MoE architecture, highlighting their structural similarities. Building upon existing approaches, the proposed framework integrates sparse MoE layers into the feed-forward layer of the encoder and the attention output projection layer of the decoder, while incorporating the preference vector into the input of the gating function.

**Strengths:**

1. This paper validates the effectiveness of incorporating MoE layers into the feed-forward layer of the encoder and the attention output projection layer of the decoder. I think this is the primary technical contribution of the work, as it can contribute to the further development of foundation models for general MOCO problems.

2. The paper presents extensive experimental results, including comparisons with prior methods, ablation studies, and hyperparameter analyses.

**Weaknesses:**

1. In Section 4.1, the authors analyze the similarity between the attention layer and the MoE architecture. However, the attention layer and token-level MoE are fundamentally different, even though both adopt a soft routing mechanism. In the attention layer, weights are assigned to value vectors, and each value vector corresponds to a different node embedding. If each value matrix is regarded as an expert, then the inputs to these experts differ. In contrast, in token-level MoE, the weights are assigned to the outputs of experts that receive the same node embedding as input. Moreover, attention layers and instance-level MoE are also distinct. Although the attention layer exhibits some structural similarity to the MoE layer and can loosely be interpreted as a form of MoE, they remain essentially different in nature.

2. While this paper highlights the similarity between the attention layer and the MoE layer, it does not provide a theoretical explanation for why MoE should outperform non-MoE architectures. Therefore, it is unclear why this similarity is emphasized or how it substantiates the paper’s contribution. The current analysis does not convincingly support the claimed benefits of the proposed approach.

3. The authors integrate MoE layers into the feed-forward layer and the attention output projection layer. Although these integrations yield empirical improvements, they are not clearly motivated by the earlier analysis of the relationship between attention and MoE layers. As a result, these sections appear disconnected and lack a coherent narrative.

**Questions:**

1. Could the authors clarify the meaning of “shared capacity” mentioned in line 150?

2. How does the statement that “the experts are token-indexed linear maps tied to the value projection” lead to the conclusion that “the utilization of experts cannot be explicitly controlled” in line 195?

---

### Official Review · Reviewer_Fga8 · 2025-11-01

**Soundness:** 3
**Presentation:** 3
**Contribution:** 3
**Rating:** 4
**Confidence:** 3

**Summary:**

This paper proposes a Preference-Aware Mixture-of-Experts (PA-MoE) framework that uses a novel preference-aware gating mechanism and sparse experts to improve performance and parameter efficiency in multi-objective combinatorial optimization.

**Strengths:**

1.  It does improve efficiency compared to traditional methods.
2.  The performance is very good.

**Weaknesses:**

1.  The key highlight is efficiency, but compared to WE-CA-aug64, the improvement is limited; in fact, looking at Table 1, it's even less efficient.
2.  The overall performance improvement is marginal.

**Questions:**

N/A

---

### Official Review · Reviewer_iHxf · 2025-11-03

**Soundness:** 3
**Presentation:** 3
**Contribution:** 3
**Rating:** 6
**Confidence:** 3

**Summary:**

The authors' core contribution stems from a novel theoretical insight: they reinterpret the current "single-model" paradigm (which uses a single, preference-conditioned Transformer) as an "implicit" Mixture-of-Experts model. To address this, the proposed PA-MoE framework replaces standard feed-forward layers with explicit, sparsely activated expert modules. The key innovation is a lightweight, "preference-aware" gating mechanism that intelligently routes inputs to experts based on both the problem instance features and the user's preference vector.

**Strengths:**

● Paper is very well written and easy to understand
● The re-interpretation of preference-conditioned attention as an "implicit, dense MoE" is a novel contribution and the proposed solution directly and elegantly follows from the theoretical motivation
● Show the robustness of their framework by successfully integrating it into three separate SOTA backbones (PMOCO, CNH, WE-CA) and showing performance gains across all of them.
● High quality ablation studies validating the design choices

**Weaknesses:**

● The top-k value was consistently set to K=2.. How was this value chosen? Have you analyzed the model's sensitivity to K? For example, would K=1 force more aggressive specialization and improve performance, or would it lead to training collapse?

**Questions:**

● The additive gating mechanism is linear and simple. Did you experiment with more complex, non-linear interaction functions (e.g., a small MLP) to combine the instance and preference embeddings and make routing decisions?

---

### Note · Authors · 2025-12-04

I have read and agree with the venue's withdrawal policy on behalf of myself and my co-authors.